# The Effects of Dietary Pattern on Metabolic Syndrome in Jiangsu Province of China: Based on a Nutrition and Diet Investigation Project in Jiangsu Province

**DOI:** 10.3390/nu13124451

**Published:** 2021-12-13

**Authors:** Yuanyuan Wang, Yue Dai, Ting Tian, Jingxian Zhang, Wei Xie, Da Pan, Dengfeng Xu, Yifei Lu, Shaokang Wang, Hui Xia, Guiju Sun

**Affiliations:** 1Key Laboratory of Environmental Medicine and Engineering of Ministry of Education, and Department of Nutrition and Food Hygiene, School of Public Health, Southeast University, Nanjing 210009, China; wyypro@foxmail.com (Y.W.); 18915999341@163.com (Y.D.); pantianqi92@foxmail.com (D.P.); withxu@seu.edu.cn (D.X.); luyifei3377@163.com (Y.L.); shaokangwang@seu.edu.cn (S.W.); huixia@seu.edu.cn (H.X.); 2Institute of Food Safety and Assessment, Jiangsu Provincial Center for Disease Control and Prevention, Nanjing 210009, China; tianting5796@126.com (T.T.); z18252063009@163.com (J.Z.); jscdcxiewei@sina.com (W.X.)

**Keywords:** dietary pattern, metabolic syndrome, factor analysis, Jiangsu Province

## Abstract

Metabolic syndrome, a complex group of metabolic disorders of energy use and storage, is considered as an important determinant risk factor for many cardiovascular diseases. This study aimed to examine the association between metabolic syndrome (MetS) and dietary pattern among adults in Jiangsu Province of China. Data were from three rounds of cross–sectional nutrition and diet investigation projects in Jiangsu Province of China, which were conducted in 2002, 2007, and 2014 by Jiangsu Provincial Center for Disease Control and Prevention. A total of 13,944 participants with complete food frequency questionnaire (FFQ) were eventually included in this study after further data screening. The 2009 Joint Interim Statement for China was used to define metabolic syndrome. Three distinct dietary patterns were identified by factor analysis: the modern dietary pattern (rich in pork, poultry, vegetables, seafood, pastry food, other animal meats, fruits, milk and its products, soft drink, whole grains, nuts, and seeds, but low in wheat), vegetable oils/condiments/soy products dietary pattern (rich in vegetable oils, other condiments, salt, soy products, and fruits and low in dry legumes), and modern high–wheat dietary pattern (rich in wheat, tubers, fruits, and other animal meats, but low in rice). Higher intake of the modern dietary pattern and modern high–wheat dietary pattern were positively associated with metabolic syndrome in both unadjusted and adjusted models by genders, whereas higher intake of the vegetable oils/condiments/soy products dietary pattern had a negative relationship with metabolic syndrome in both unadjusted and adjusted models by genders (*p* < 0.05). Our study recommends reducing the consumption of animal meat products, especially processed meat products, and replacing animal oils with vegetable oils as the main supply of daily oils. Furthermore, more prospective and experimental studies are needed to confirm the relationship between dietary patterns and metabolic syndrome.

## 1. Introduction

Metabolic syndrome (MetS), also known as “Syndrome X” and “Insulin Resistance Syndrome”, is a complex group of metabolic disorders of energy use and storage [1]. This syndrome, characterized by centripetal obesity, dyslipidemia, increased blood pressure, and elevated blood glucose levels, is one of the most serious non–communicable chronic diseases and increases the risk of type 2 diabetes and cardiovascular disease (CVD) [2,3,4]. Unfortunately, regardless of the diagnostic criteria used, the prevalence of metabolic syndrome increases year on year worldwide. From 2011 to 2016, the overall prevalence of metabolic syndrome in the United States was 34.7% (95% CI, 33.1–34.3%), and the prevalence was not significantly different among men and women (35.1% vs. 34.3%; *p*  = 0.47) [5]. A systematic review and meta–analysis of 226,653 Chinese people showed that 24.5% (95% confidence interval (CI): 22.0–26.9%) had metabolic syndrome, with a prevalence of 19.2% in men and 27.0% in women [6]. In addition, the high prevalence of metabolic syndrome in Chinese children is not negligible, rising from 2.3% in 2004 to 3.2% in 2014 [7]. This is an extremely grim figure and bodes well for the future of metabolic syndrome in our country as an increasingly tough health problem. Epidemiological studies showed that environmental factors such as socio–economic status, sedentary lifestyle, and diet are significantly related to the occurrence of metabolic syndrome [4,7]. Among them, diet as a controllable environmental factor plays an important role in the increased prevalence of metabolic syndrome.

The importance of single diet or dietary composition on metabolic syndrome has been assessed [8,9,10]. In current nutritional epidemiological investigations, dietary patterns are more frequently used to assess the complex interactions of nutrients and foods with various diseases and their synergistic effects on the same diseases. Dietary patterns are more representative of the overall dietary habits of the study population. Therefore, examining dietary patterns may be a better predictor of disease risk than individual nutrients or foods [11,12,13]. A systematic review and meta–analysis including 40 observational studies showed that adherence to a “healthy” dietary pattern—characterized as rich in vegetables and fruit, poultry, fish, and whole grains—was associated with a reduced risk of metabolic syndrome, and was more pronounced in men and Asian countries. However, adherence to a “meat/Western” pattern (which is characterized by a high loading of red meat, processed meat, animal fat, eggs, and sweets) increased the risk of metabolic syndrome in stratified analyses based on either geographical region (Asia, Europe, America) or study design [14]. A previous study found that a Mediterranean diet, which is characterized as rich in vegetables, fruit, whole grains, cereal, fish, and seafood products, was associated with a lower frequency of metabolic syndrome and reduced all–cause mortality [15,16].

Recent decades have witnessed the rapid economic development of Jiangsu Province. This rapid economic development has then accelerated the change in the region’s dietary habits and disease profile. The prevalence of metabolic syndrome in Jiangsu Province of China has been reported to be increasing year by year, but few articles have been found to assess the relationship between dietary pattern and metabolic syndrome in this region [17,18,19]. Therefore, the purpose of this study was to examine the association between dietary pattern and metabolic syndrome among adults in Jiangsu Province of China.

## 2. Methods

### 2.1. Study Population 

Data were from three rounds of cross–sectional nutrition and diet investigation projects in Jiangsu Province of China, which were conducted in 2002, 2007, and 2014 by Jiangsu Provincial Center for Disease Control and Prevention. The three–round surveys used the same multi–stage stratified cluster random sampling method. The screening process for survey respondents is shown in Figure 1. Of the total 44,525 respondents, those aged 18 years and over with a completed food frequency questionnaire (FFQ) were included in the study (n = 23,677). Then, we excluded data from those with abnormal energy intake (<500 or ≥4000 kcal per day, n = 5594). Among the remaining respondents, we further excluded people with chronic diseases such as diabetes and stroke, as their diet may be altered owing to their chronic diseases (n = 999). Individuals with insufficient information on body composition, blood pressure, lipid levels, and fasting plasma glucose (FPG) were further excluded from this study (n = 3090). A total of 13,944 participants (5954 men and 7990 women) were finally included in this study. This study was approved by the Ethics Committee of the Jiangsu Provincial Center for Disease Control and Prevention, reference number JSCDC2014236. In addition, all participants signed an informed consent form for this study.

### 2.2. Demographic and Lifestyle Survey 

Demographic data including age, gender, education level (primary and below, secondary, and senior secondary and above), physical work (low physical work, middle physical work, high physical work, and other work), region (city and rural), and geographical region (divided into southern and northern Jiangsu Province according to geographical location) were surveyed by professionally trained investigators. Smoking/drinking status was defined as who had the habits of smoking and/or drinking during the investigation [18].

### 2.3. Anthropometric Measurements

All anthropometric measurements were carried out in a comfortable environment with light clothing. Weight was measured in kilogram (kg). Height and waist circumference was measured in centimeter (cm) [20]. All measurements were performed twice using a standard protocol and techniques. Body mass index (BMI) was calculated as weight (kg) divided by square of height (meter (m)) [21].

### 2.4. Blood Pressure Measurement and Biochemical Indicator

Blood pressure was likewise measured in a quiet and comfortable environment. The blood pressure of all respondents was measured three times by a professional investigator and the final result was averaged from the three measurements [22]. All blood samples were collected in the morning after fasting during the night. All samples were subjected to rigorous quality control analysis by the Jiangsu Centre for Disease Control and Prevention, which measured the concentrations of triglyceride (TG), high density lipoprotein cholesterol (HDL), and FPG.

### 2.5. Definition of Metabolic Syndrome

The metabolic syndrome as a collective term for a group of risk factors has not been defined uniformly for more than a decade, and in this survey, we mainly used the definition of metabolic syndrome from the 2009 Joint Interim Statement for China [23]: (1) elevated waist circumference: waist circumference ≥85 cm in males, ≥80 cm in females; (2) elevated TG: TG >1.70 mmol/L (150 mg/dL) or those who have been treated for high TG; (3) reduced HDL: HDL <1.0 mmol/L (40 mg/dL) in males, <1.30 mmol/L (50 mg/dL) in females or those who have been treated for low HDL; (4) elevated blood pressure: systolic blood pressure (SBP) ≥130 and/or diastolic blood pressure (DBP) ≥85 mmHg or those who were treated for hypertension; (5) elevated FPG: FPG ≥5.6 mmol/L (100 mg/dL) or those who received anti–hyperglycemia treatment. Among the five points above, metabolic syndrome can be confirmed if three or more of them were met. 

### 2.6. Dietary Assessment

All dietary information was collected by a standardized semi–quantitative food frequency questionnaire (FFQ) [24]. The FFQ contained hundreds of kinds of food, which basically covered dietary intake of residents in Jiangsu Province for one year. The food frequency questionnaire included all the foods consumed daily by residents of Jiangsu Province. The food model was used to allow the investigators to recall the amount of foods eaten. At the same time, participants were asked to recount the frequency of personal food consumption (number of times per day, week, month, and year) as well as the number of servings. Ultimately, food intake was transformed into g/day for data analysis. Then, all foods were divided into 23 food groups according to the Chinese Dietary Guidelines and the available explanations, which represent the dietary habits of Jiangsu residents in the last decade or so, as shown in Table 1 [25].

### 2.7. Statistical Analysis

Factor analysis (FA) was performed to infer dietary patterns in different genders. Orthogonal rotations using Kaiser’s criterion (eigenvalues > 1.3) were utilized to establish the number of factors. Only the absolute values of factor loading >0.26 were included in the analysis, as these items represented the foods with the strongest relationships to the identified factors. The dietary patterns were named according to the magnitude of the factor scores and the interpretability of the overall diet. Factor scores were further divided into three quartiles. Participants were divided into three groups based on the dietary pattern obtained. All categorical variables were expressed using numbers (percentages), and continuous variables were expressed using means ± standard deviations (SD). Chi–square tests and generalized linear models were used to explore differences in the general characteristics of participants. Multivariable logistic regression analysis was used to compare the odds ratios (ORs) and 95%CI to analyze the relationship between dietary patterns and metabolic syndrome by gender. Data management and statistical analysis were performed using IBM SPSS Statistics software Version 26.0 and OriginLab (2021) Data analysis and graphing software. All analyses were defined as statistically different at *p*–values (two sides) <0.05.

## 3. Results

### 3.1. Determination of Dietary Patterns

Dietary patterns from different genders were identified by factor analysis. Both the Kaiser–Meyer–Olkin index (0.704 in males and 0.705 in females) and Bartlett’s test (*p* < 0.001) showed that the sample was suitable for FA.

Among males, finally, three dietary patterns were established and named as the modern dietary pattern (pattern I), vegetable oils/condiments/soy products dietary pattern (pattern II), and modern high–wheat pattern dietary pattern (pattern III) according to the highest factor loading of food items and interpretability, as shown in Figure 2a and Table 2. The modern dietary pattern (pattern I), vegetable oil/condiment/soy products dietary pattern (pattern II), and modern high–wheat pattern dietary pattern (pattern III) can explain 10.732%, 10.310%, and 8.149% of the variance, respectively.

Among females, three similar dietary patterns were finally identified and named as the modern dietary pattern (pattern I), vegetable oils/condiments/soy products dietary pattern (pattern II), and modern high–wheat pattern dietary pattern (pattern III), which can be seen in Figure 2b and Table 2. The modern dietary pattern (pattern I), vegetable oil/condiment/soy products dietary pattern (pattern II), and modern high–wheat pattern dietary pattern (pattern III) can explain 10.766%, 10.049%, and 8.537% of the variance, respectively.

### 3.2. Basic Information of Participants 

Of the 13,944 respondents, 28.5% (95%CI: 27.8–29.2%) had metabolic syndrome, with a significantly higher prevalence among women (36.1% (95%CI: 30.6–32.6%)) than men (24.4% (95%CI: 23.3–25.5%)) (*p* < 0.001). The average of waist circumstance in men was 83.0 ± 9.7 cm, which was higher than that in women (79.8 ± 10.0 cm) (*p* < 0.001). In addition, smoking and alcohol consumption rates were higher in men than those in women, with statistically significant differences (*p* < 0.001) (table not shown).

### 3.3. Characteristics of the Participants in Dietary Patterns

General characteristics of participants across the three dietary patterns across tertiles are shown in Table 3. 

Among males, participants in higher tertile of modern dietary pattern (pattern I) were more likely to be younger and more concentrated in the southern Jiangsu Province (*p* < 0.001); had a greater portion of participants who were unmarried; had higher education level and lower physical work; and had higher energy intake, BMI level, waist circumference, TG, and FPG compared with participants in the lowest quartile of pattern I (*p* < 0.05), while participants from higher quartile of the vegetable oils/condiments/soy products dietary pattern (pattern II) were more likely to be younger and more concentrated in urban areas and the southern Jiangsu Province (*p* < 0.001); had smoking and drinking behavior; and had higher energy intake and DBP level, but lower waist circumference, SBP, TG, HDL, and FPG levels compared with the participants in the lowest quartile of pattern II (*p* < 0.05). Meanwhile, participants in higher quartile of the modern high–wheat dietary pattern (pattern III) were likely to be younger and more concentrated in northern Jiangsu Province (*p* < 0.05); had a greater portion of individuals who had lower education levels and lower physical work; had lower rates of smoking and drinking; and had higher BMI level, waist circumference, DBP, and FPG levels, but lower energy intake compared with the participants in the lowest quartile of pattern III (*p* < 0.001).

Among females, individuals in higher tertile of the modern dietary pattern (pattern I) were more likely to be younger and more concentrated in rural areas and the southern Jiangsu Province (*p* < 0.001); had a greater portion of participants who were married; had a higher education level and lower physical work; and had higher energy intake, BMI, waist circumference, TG, and FPG levels, but lower SBP, DBP, and HDL levels compared with participants in the lowest quartile of pattern I (*p* < 0.05), while participants from higher quartile of the vegetable oils/condiments/soy products dietary pattern (pattern II) were more likely to be younger and more concentrated in urban areas and northern Jiangsu Province (*p* < 0.001); had smoking and drinking behavior; and had higher energy intake, BMI, and DBP levels, but lower SBP, TG, HDL, and FPG levels compared with the participants in the lowest quartile of pattern II (*p* < 0.05). Meanwhile, participants in higher quartile of the modern high–wheat dietary pattern (pattern III) were likely to be younger and more concentrated in urban areas and the northern Jiangsu Province (*p* < 0.001); had a greater portion of individuals who had higher education levels and lower physical work; had lower rates of smoking and drinking; and had higher waist circumference, BMI, DBP, and FPG levels, but lower SBP level compared with the participants in the lowest quartile of pattern III (*p* < 0.05).

### 3.4. Association between Dietary Patterns and Metabolic Syndrome by Gender

The results of the relationship between dietary patterns and metabolic syndrome in different genders using multivariate logistic regression are displayed in Table 4. Among males, participants in the higher tertiles of the modern dietary pattern (pattern I) and modern high–wheat pattern dietary pattern (pattern III) have a positive influence on metabolic syndrome in both unadjusted model and adjusted model (composed to T1, OR = 1.530, 95% CI: 1.271–1.842 in pattern I; OR = 1.360, 95% CI, 1.172–1.578 in pattern III, respectively, *p* < 0.05). On the other hand, the higher intake of vegetable oils/condiments/soy products dietary pattern score was negatively associated with metabolic syndrome in both unadjusted model and adjusted model (compared with T1, OR = 0.692, 95% CI: 0.598–0.800, *p* < 0.05). Similar to males, both the modern dietary pattern (pattern I) and modern high–wheat pattern dietary pattern (pattern III) with higher scores among females were positively associated with metabolic syndrome, but the difference was that such a positive association was only seen in the adjusted model (compared with T1, OR = 1.289, 95% CI: 1.104–1.505 in pattern I; OR = 1.203, 95% CI, 1.030–1.406 in pattern III, respectively, *p* < 0.05). Conversely, participants with a higher score of vegetable oils/condiments/soy products dietary pattern were negatively correlated with metabolic syndrome in both unadjusted model and adjusted model (compared with T1, OR = 0.863, 95% CI: 0.746–0.977, *p* < 0.05).

## 4. Discussion

Metabolic syndrome is a cluster of risk factors that increases the risk of type 2 diabetes by five times and cardiovascular disease by three times, and has unfortunately become a global public health problem [2]. Diet, as an important part of lifestyle, has been demonstrated to be significantly associated with metabolic syndrome [26,27]. In this cross–sectional study, three dietary patterns including the modern dietary pattern, vegetable oils/condiments/soy products dietary pattern, and modern high–wheat dietary pattern were established using factor analysis to analyze the relationship between diet and metabolic syndrome.

The modern dietary pattern, which was high in pork, poultry, vegetables, seafood, pastry food, other animal meats, fruits, milk and its products, soft drink, whole grains, nuts, and seeds, but low in wheat, was found to have a positive association with metabolic syndrome in both men and women. This dietary pattern was similar to the modern dietary pattern obtained previously from China Health and Nutrition Survey (CHNS), which was found to be positively associated with general and central obesity [28,29]. In addition, the modern dietary pattern in our study contained a high consumption of animal meats, including processed meat and red meat, which were consistent with the Western dietary pattern. The Western dietary pattern, which is rich in red meat, processed meat, pasta, and rice, has been found to promote inflammatory responses and CVDs [30]. Data from the cohort study indicated that the Western dietary pattern was negatively associated with HDL cholesterol and positively associated with all other response variables (LDL–cholesterol, systolic and diastolic blood pressure, fasting glucose, and insulin) [27]. Soft drinks including sugar–sweetened beverages (SSBs) and artificially sweetened beverages (ASBs) were also the main components of modern dietary pattern in our study. A systematic review and meta–analysis demonstrated that the high consumption of SSBs and ASBs was positively associated with metabolic syndrome [31]. Besides, the increased intake of ASBs and SSBs also contributed to the increased risk of CVD death and all–cause mortality [32]. Energy provided by sugar is highly likely to cause weight gain if it is not eventually metabolized and balanced by physical activity. Numerous epidemiological surveys have found that obesity was a major risk factor for insulin resistance, type 2 diabetes, dyslipidemia, and hypertension, which are components of metabolic syndrome [2,13]. Although the dangers of soft drinks have reached a point where we have to be aware of them, the market for soft drinks in China is still booming. A CHNS survey found that the beverage consumption prevalence rate rose from 22.4% in 2004 to 33.1% in 2011 [33]. Moreover, the consumption of pastry snacks rich in high energy density was another important components of modern dietary patterns. Despite the general interest in the idea of consuming snacks, ready–to–eat, highly processed snacks are becoming more readily available and consumed. Such highly processed snacks tend to be higher in energy, which leads to an excessive accumulation of energy in the body and, ultimately, to obesity or other metabolic diseases [34]. Therefore, it is important to raise concerns about reducing the public’s obsession with soft drinks and pastry snacks. Notably, the modern dietary pattern also contains some beneficial food components, such as fish, vegetables, whole grains, and fruits. A systematic review and meta–analysis demonstrated that fish consumption might have a preventive role in the development of metabolic syndrome only in men [35]. Some studies hold the opposite opinion that people with metabolic syndrome might have a higher intake of protein, meat, and fish [36]. In addition, a study from Hangzhou of China showed that adherence to a seafood diet had no association with the risk of metabolic syndrome [37]. The relationship between various seafood products, such as fish, and metabolic syndrome is still confusing. Hence, it is necessary to conduct further studies to determine the ability of fish and long–chain n–3 fatty acid depletion to improve metabolic syndrome and its composition. Numerous clinical studies have demonstrated the beneficial effects of vegetables and fruits on health [38,39]. The importance of fruits and vegetables was reflected in various dietary guidelines [25,40]. However, in our study, vegetables and fruits did not provide sufficient protection against metabolic syndrome. This might be due to the high intake of meat in modern dietary patterns, which offset the beneficial effects of fruits and vegetables.

The modern high–wheat dietary pattern, which was high in wheat, tubers, fruits, and other animal meats, but low in rice (listed here are only the parts that are the same for both genders), was more common in northern Jiangsu Province. This dietary pattern was more inclined to the integration of the traditional Northern dietary pattern and Western dietary pattern [41,42,43]. Zuo Hui, et al. noted that the adherence of people in Jiangsu region to a high–wheat dietary pattern led to a positive association with insulin resistance, which is thought to be the underlying mechanism for metabolic syndrome after adjustment for age and sex. Further, the positive association remained after adjustment for income, total energy intake, and physical activity, although it disappeared after an additional increase in BMI [44]. Notably, an interesting observation was that the modern dietary pattern, where men and women consumed less wheat, was positively associated with metabolic syndrome, while the modern high–wheat dietary pattern characterized by a high intake of wheat and a low intake of rice was still positively associated with metabolic syndrome. The main reason for this phenomenon might be that it was not just rice or wheat alone that was responsible for the development of metabolic syndrome. This result might be best explained by the focus on the overall diet rather than the impact of a single food or nutrient on health. The high consumption of milk and its products was contained in this dietary pattern. In a systematic review and meta–analysis, the results have shown that higher intake of dairy products significantly reduces the risk of metabolic syndrome by 17% in cross–sectional studies and by 14% in cohort studies [45]. However, the other study held the opposite opinion that the more cheese the elderly eat, the higher the risk of metabolic syndrome [46]. However, in our study, all kinds of milk and dairy products were grouped together into one food subgroup, so it remained controversial whether they were beneficial. Therefore, subsequent studies should consider separating the different types of dairy products to explore the relationship with metabolic syndrome. Fewer articles have been written on the relationship between eggs and the metabolic syndrome. A systematic review and meta–analysis of a prospective study suggested that egg intake was associated with an increased incidence of type 2 diabetes in the general population and an increased incidence of CVD co–morbidity in patients with diabetes [47]. On the contrary, data from Korean health examinees study showed that a higher weekly intake of eggs was associated with a reduced risk of metabolic syndrome and its components in women [48]. Therefore, the protective effect of eggs remains to be proven by more studies.

To our surprise, the vegetable oils/condiments/soy products dietary pattern, dominated by a high score of vegetable oils, other condiments, salt, soy products, and fruits and low in dry legumes, was found to be negatively correlated with metabolic syndrome. Salty foods act as ‘addictive substances’, stimulating the brain’s reward and pleasure mechanisms to crave salty foods, which may also be responsible for the increased incidence of obesity–related diseases [49]. In addition, a high salt intake has been reported as an independent risk factor for hypertension and might likewise increase the incidence of metabolic syndrome in healthy people [50,51,52]. Therefore, we used salt as a separate food grouping to examine whether salt also increased the prevalence of metabolic syndrome in the whole diet at the beginning of the study. The result was surprising as the factor scores for salt and other condiments were higher in the vegetable oil/condiment/soy products dietary pattern, whereas this pattern showed a protective effect against metabolic syndrome in both the unadjusted model and the model adjusted for genders. The reason for this result may be that other components of this dietary pattern have the opposite effect, e.g., vegetable oil. Different from animal oils containing large amounts of saturated fatty acids, vegetable oils contain high levels of unsaturated fatty acids and have been reported in numerous studies to improve lipocalin concentrations, insulin resistance, and body composition in people by increasing their levels in foods, even without changing other components of the diet or fat intake [53,54,55]. Most observational studies have found a positive association between monounsaturated fatty acids’ (MUFAs) and polyunsaturated fatty acids’ (PUFAs) (both n–3 and n–6 subtypes) intake and metabolic syndrome components. The benefits of diets rich in MUFA or PUFA, including low–fat diets, in reducing metabolic syndrome were also supported by clinical trials [56,57,58]. Another major reason was the higher intake of soy products in this dietary pattern. Research has found that the regular intake of cultivated soy foods, which are unique to Asia, appeared to be associated with a lower prevalence of metabolic syndrome [59,60]. In addition, soy intake tends to have a gender–dependent effect on the risk of metabolic syndrome and specific cancers [61,62]. The consumption of fruits and eggs (only in men) has been described in the above two dietary pattern discussions as beneficial for metabolic syndrome. Therefore, it is reasonable to assume that this dietary pattern was negatively associated with metabolic syndrome. However, more research is probably needed to prove it.

To the best of our knowledge, this was the first time that such a large sample size from three rounds of cross–sectional nutrition surveys had been used to study the relationship between dietary patterns and metabolic syndrome using multivariate logistic regression in Jiangsu Province of China. Second, all three rounds of the nutrition survey had a rigorous experimental design, with participant data including demographic information, lifestyle, dietary intake, anthropometric data, and biochemical measurements, providing a variety of data information and homogenization of data analysis. However, the present investigation also had some weaknesses. Firstly, because of the cross–sectional nature, we are unable to understand the causal relationship between dietary patterns by gender and metabolic syndrome. Secondly, selection bias may be unavoidable as we have removed no FFQs or other data that lead to incomplete data. Thirdly, factor analysis is an a posteriori methodological approach that relies on learned knowledge and experience in grouping foods, but the final presentation of the data does not rely on any prior knowledge, and thus the reproducibility and validity of the data are poor. Finally, although we included the physical work, the physical activities of residents were not included, which may lead to bias.

## 5. Conclusions

Three dietary patterns were obtained through factor analysis in both males and females in Jiangsu Province of China: the modern dietary pattern, the vegetable oils/condiments/soy products dietary pattern, and the modern high–wheat dietary pattern. The modern dietary pattern and modern high–wheat dietary pattern had a positive association with metabolic syndrome in both males and females. The vegetable oils/condiments/soy products dietary pattern was negatively associated with metabolic syndrome in both males and females. Our study recommends that the consumption of animal meat products, especially processed meat products, should be reduced and that vegetable oils should replace animal oils as the main supply of daily oil in Jiangsu Province of China. Furthermore, more prospective and experimental studies are needed to confirm the relationship between dietary patterns and metabolic syndrome.

## Figures and Tables

**Figure 1 nutrients-13-04451-f001:**
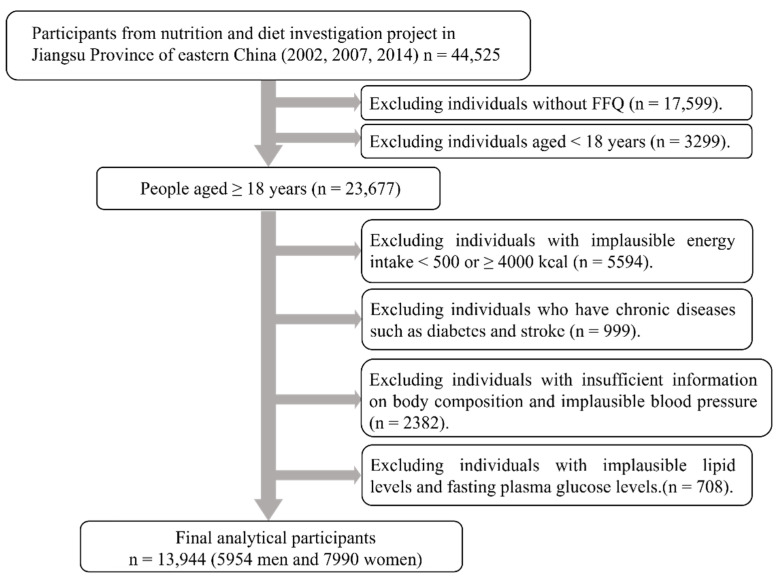
Flowchart of the selection of research participants from three rounds of cross–sectional nutrition and diet investigation projects. 2002, n = 2295; 2007, n = 6700; 2014, n = 4949.

**Figure 2 nutrients-13-04451-f002:**
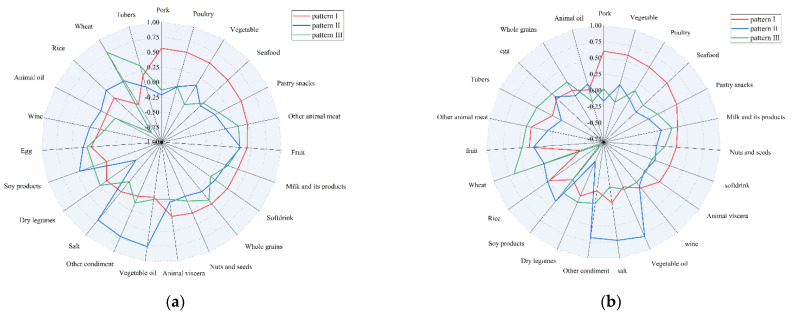
Radar chart of different dietary patterns from factor analysis. (**a**) radar chart in males; (**b**) radar chart in females.

**Table 1 nutrients-13-04451-t001:** Food groupings used in factor analysis.

Food Group	Example of the Food Group
Rice	Rice, rice flour
Wheat	noodles, pasta, plain bread
Whole grains	Barley, buckwheat, millet, corn
Tubers	Sweet potato, potato, Chinese yam, taro
Vegetables	Spinach, canola, carrot, spinach, preserved vegetables
Soy products	Soybeans, soymilk, tofu
Dry legumes	Black beans, red lentils, kidney beans, green beans
Fruits	Fresh and canned (no added sugar) fruits
Eggs	Whole eggs, yolk, white, preserved eggs
Milk and its products	Whole milk, skim milk, flavored milk, cheese, yogurt
Poultry	Chicken, duck meat
Pork	pork and its products
Other animal meat	beef, lamb, and those products
Animal viscera	Viscera products of animals
Seafood	Fresh fish, dried fish, shellfish, shrimp
Nuts and seeds	Sesame, sunflower, peanuts, walnuts, almonds, hazelnuts, pine–nuts
Vegetable oils	Soybean oil, peanut oil, sesame oil
Animal oils	Butter, lard, sheep oil
Salt	Salt
Other condiments	Sauce, soy sauce, monosodium glutamate
Wine	Beer, rice wine, white wine
Soft drink	Fruit or flavored drinks, fruit juice, soft drinks
Pastry snacks	Cakes, pancake, mooncake

**Table 2 nutrients-13-04451-t002:** Factor loadings and dietary patterns for the 23 food groups derived from factor analysis.

Food Groups	Men			Women		
	Pattern I	Pattern II	Pattern III	Pattern I	Pattern II	Pattern III
Pork	0.564			0.603		
Poultry	0.556			0.553		
Vegetables	0.544		−0.266	0.592		
Seafood	0.512			0.535		
Pastry snacks	0.494			0.466		
Other animal meats	0.462		0.310	0.341		0.414
Fruits	0.429	0.319	0.306	0.347	0.279	0.458
Milk and its products	0.348			0.362		0.275
Soft drink	0.345			0.279		
Whole grains	0.325					0.286
Nuts and seeds	0.290			0.322		
Animal viscera						
Vegetable oils		0.758			0.784	
Other condiments		0.717			0.684	
Salt		0.673			0.727	
Dry legumes		−0.477			−0.479	
Soy products		0.450			0.374	
Eggs		0.309				0.291
Wine						
Animal oils						
Rice		0.262	−0.773			−0.739
Wheat	−0.266		0.753	–0.418		0.666
Tubers			0.322			0.359

Factor loadings of <0.26 in absolute terms were excluded for simplicity; pattern I: modern dietary pattern; pattern II: vegetable oils/condiments/soy products dietary pattern; pattern III: modern high–wheat pattern dietary pattern.

**Table 3 nutrients-13-04451-t003:** General characteristics of the participants under the three dietary patterns by genders.

Tertile of Each Pattern Score								
Dietary Pattern	Men				Women			
	T1	T2	T3	*p* Value	T1	T2	T3	*p* Value
Pattern I								
Age (years)	54.4 ± 15.2 ^a^	52.3 ± 14.9	49.0 ± 15.4	<0.001	52.7 ± 16.0 ^a^	50.6 ± 14.8	48.4 ± 14.5	<0.001
BMI (kg/m^2^)	23.4 ± 3.3 ^a^	23.6 ± 3.2	24.0 ± 3.3	<0.001	24.0 ± 3.6 ^a^	23.7 ± 3.5	23.8 ± 3.6	0.007
Energy intake (kcal/d)	2189.5 ± 618.9 ^c^	2228.4 ± 653.2	2454.5 ± 683.0	<0.001	1939.0 ± 556.4 ^c^	1968.1 ± 563.8	2222.3 ± 598.8	<0.001
Waist circumference (cm)	81.6 ± 9.6 ^a^	82.8 ± 9.5	84.6 ± 9.8	<0.001	80.4 ± 10.2 ^a^	79.4 ± 9.9	79.7 ± 10.0	<0.001
SBP (mm Hg)	129.3 ± 20.1	129.7 ± 19.0	128.3 ± 17.6	0.061	127.3 ± 22.3 ^a^	126.2 ± 20.6	124.6 ± 19.7	<0.001
DBP (mm Hg)	82.1 ± 11.4 ^b^	82.8 ± 10.5	82.6 ± 10.4	0.088	79.9 ± 11.9 ^c^	79.4 ± 10.2	78.6 ± 10.2	<0.001
TG (mmol/L)	1.4 ± 1.5 ^a^	1.6 ± 1.7	1.9 ± 2.0	<0.001	1.4 ± 1.1 ^a^	1.5 ± 1.2	1.6 ± 1.3	<0.001
HDL–C (mmol/L)	1.3 ± 0.4	1.3 ± 0.4	1.3 ± 0.4	0.879	1.3 ± 0.3 ^a^	1.3 ± 0.3	1.3 ± 0.4	<0.001
FPG (mmol/L)	5.0 ± 1.1 ^a^	5.1 ± 1.3	5.1 ± 1.2	<0.001	5.0 ± 1.2	5.0 ± 1.1	5.1 ± 1.1	0.019
Education level				<0.001				<0.001
Primary school or less	1130 (47.7) ^c^	947 (40.0)	770 (32.5)		1834 (68.8) ^c^	1478 (55.5)	1168 (43.9)	
Junior high school	891 (37.6)	981 (41.4)	1027 (43.4)		696 (26.1) ^c^	850 (31.9)	972 (36.5)	
High school and higher	349 (14.7) ^c^	441 (18.6)	572 (24.1)		134 (5.0) ^c^	335 (12.6)	523 (19.6)	
Physical work				<0.001				<0.001
Low physical work	806 (40.6) ^c^	968 (48.8)	1111 (55.9)		1536 (57.7)	1508 (56.6) ^c^	1669 (62.7)	
Middle physical work	119 (6.0) ^a^	228 (11.5)	237 (11.9)		57 (2.1) ^c^	154 (5.8)	186 (7.0)	
High physical work	780 (39.3) ^a^	462 (23.3)	344 (17.3)		876 (32.9) ^c^	679 (25.5)	507 (19.0)	
Other physical work	279 (14.1)	326 (16.4)	294 (14.8)		195 (7.3) ^c^	322 (12.1)	301 (11.3)	
Region				0.278				<0.001
City	610 (30.7)	564 (28.4)	589 (29.7)		812 (30.5)	916 (34.4)	755 (28.4)	
Rural	1374 (69.3)	1420 (71.6)	1397 (70.3)		1852 (69.5) ^b^	1747 (65.6)	1908 (71.6)	
Geographical region				<0.001				<0.001
Southern Jiangsu	672 (33.9) ^a^	1347 (67.9)	1447 (72.9)		682 (25.6) ^a^	1676 (62.9)	1926 (72.3)	
Northern Jiangsu	1312 (66.1)	637 (32.1)	539 (27.1)		1982 (74.4)	987 (37.1)	737 (27.7)	
Marital status				<0.001				<0.001
Unmarried	71 (3.6) ^c^	79 (4.0)	114 (5.7)		60 (2.3) ^c^	58 (2.2)	90 (3.4)	
Married	1791 (90.3)	1813 (91.4)	1812 (91.2)		2253 (84.6) ^c^	2356 (88.5)	2394 (89.9)	
Divorced	16 (0.8)	13 (0.7)	14 (0.7)		23 (0.9)	20 (0.8)	25 (0.9)	
Widowed	106 (5.3) ^c^	79 (4.0)	46 (2.3)		328 (12.3) ^c^	229 (8.6)	154 (5.8)	
Smoking behavior				0.007				0.126
No	974 (49.1)	887 (44.7)	972 (48.9)		2596 (97.4)	2609 (98.0)	2616 (98.2)	
Yes	1010 (50.9) ^b^	1097 (55.3)	1014 (51.1)		70 (2.6) ^c^	167 (2.0)	287 (1.8)	
Alcohol consumption				0.100				<0.001
Non–drinker	1142 (57.6)	1075 (54.2)	1113 (56.0)		2606 (97.8)	2558 (96.1)	2544 (95.5)	
Current drinker	842 (42.4)	909 (45.8)	873 (44.0)		58 (2.2) ^a^	105 (3.9)	119 (4.5)	
Pattern II								
Age (years)	54.1 ± 15.4 ^a^	52.5 ± 15.2	49.2 ± 15.0	<0.001	52.5 ± 15.8 ^a^	50.0 ± 15.4	49.2 ± 14.2	<0.001
BMI (kg/m^2^)	23.8 ± 3.2 ^c^	23.6 ± 3.3	23.5 ± 3.2	0.112	23.7 ± 3.5 ^c^	23.8 ± 3.6	24.0 ± 3.6	0.002
Energy intake (kcal/d)	1969.0 ± 716.0 ^a^	2277.1 ± 549.6	2626.2 ± 534.8	<0.001	1832.6 ± 669.5 ^a^	2033.9 ± 508.2	2262.7 ± 485.2	<0.001
Waist circumference (cm)	84.0 ± 9.9 ^a^	82.8 ± 9.6	82.4 ± 9.5	<0.001	79.9 ± 9.9	79.6 ± 10.2	79.9 ± 9.9	0.473
SBP (mm Hg)	129.8 ± 17.8 ^c^	130.0 ± 19.8	127.5 ± 19.1	<0.001	126.9 ± 20.2 ^a^	125.7 ± 21.7	125.5 ± 20.9	0.034
DBP (mm Hg)	82.1 ± 10.4 ^c^	82.7 ± 10.8	82.9 ± 11.2	0.045	79.3 ± 10.5	78.7 ± 10.9 ^c^	79.8 ± 11.1	0.002
TG (mmol/L)	1.9 ± 2.0 ^a^	1.6 ± 1.6	1.4 ± 1.5	<0.001	1.7 ± 1.4 ^a^	1.5 ± 1.2	1.3 ± 1.1	<0.001
HDL–C (mmol/L)	1.3 ± 0.4 ^a^	1.3 ± 0.4	1.2 ± 0.4	<0.001	1.3 ± 0.4 ^a^	1.3 ± 0.3	1.2 ± 0.3	<0.001
FPG (mmol/L)	5.3 ± 1.4 ^a^	5.1 ± 1.1	4.9 ± 1.1	<0.001	5.2 ± 1.0 ^a^	5.0 ± 1.0	4.9 ± 1.3	<0.001
Education level				0.001				<0.001
Primary school or less	770 (38.8)	833 (42.0) ^c^	715 (36.0)		1468 (55.1) ^c^	1456 (54.7)	1556 (58.4)	
Junior high school	816 (41.1)	795 (40.1)	837 (42.1)		813 (30.5)	864 (32.4)	841 (31.6)	
High school and higher	399 (20.1)	355 (17.9)	434 (21.9)		382 (14.3) ^c^	343 (12.9)	267 (10.0)	
Physical work				<0.001				<0.001
Low physical work	1011 (50.9)	935 (47.2)	939 (47.3)		1600 (60.1) ^b^	1502 (56.4)	1611 (60.5)	
Middle physical work	212 (10.7)	197 (9.9)	175 (8.8)		144 (5.4) ^c^	149 (5.6)	104 (3.9)	
High physical work	521 (26.2)	509 (25.7)	556 (28.0)		646 (24.3)	707 (26.5)	709 (26.6)	
Other physical work	241 (12.1) ^b^	342 (17.2)	316 (15.9)		273 (10.3)	305 (11.5)	240 (9.0)	
Region				<0.001				<0.001
City	498 (25.1) ^a^	593 (29.9)	672 (33.8)		744 (27.9) ^a^	856 (32.1)	883 (33.1)	
Rural	1487 (74.9)	1390 (70.1)	1314 (66.2)		1919 (72.1)	1807 (67.9)	1781 (66.9)	
Geographical region				<0.001				<0.001
Southern Jiangsu	1286 (64.8)	1202 (60.6)	978 (49.2)		1703 (64.0)	1411 (53.0)	1170 (43.9)	
Northern Jiangsu	699 (35.2) ^a^	781 (39.4)	1008 (50.8)		960 (36.0) ^a^	1252 (47.0)	1494 (56.1)	
Marital status				0.160				0.597
Unmarried	81 (4.1)	73 (3.7)	110 (5.5)		57 (2.1)	80 (3.0)	71 (2.7)	
Married	1813 (91.3)	1816 (91.6)	1787 (90.0)		2341 (87.9)	2322 (87.2)	2340 (87.8)	
Divorced	13 (0.7)	15 (0.8)	15 (0.8)		21 (0.8)	24 (0.9)	23 (0.9)	
Widowed	78 (3.9)	79 (4.0)	74 (3.7)		244 (9.2)	237 (8.9)	230 (8.6)	
Smoking behavior				<0.001				0.032
No	991 (49.9)	992 (50.0)	850 (42.8)		2617 (98.3)	2612 (98.1)	2592 (97.3)	
Yes	994 (50.1) ^c^	991 (50.0)	1136 (57.2)		46 (1.7) ^c^	51 (1.9)	72 (2.7)	
Alcohol consumption				<0.001				0.001
Non–drinker	1183 (59.6)	1153 (58.1)	994 (50.1)		2586 (97.1)	2582 (97.0)	2540 (95.3)	
Current drinker	802 (40.4) ^c^	830 (41.9)	992 (49.9)		77 (2.9) ^c^	81 (3.0)	124 (4.7)	
Pattern III								
Age (years)	52 ± 14.2 ^c^	52.7 ± 15.7	51.0 ± 16.0	0.002	52.2 ± 14.5 ^c^	51.4 ± 15.5	48.1 ± 15.3	<0.001
BMI (kg/m^2^)	23.4 ± 3.2 ^c^	23.5 ± 3.2	24.0 ± 3.3	<0.001	23.7 ± 3.6 ^c^	23.7 ± 3.5	24.2 ± 3.6	<0.001
Energy intake (kcal/d)	2492.1 ± 546.2 ^a^	2047.1 ± 699.1	2333.4 ± 654.4	<0.001	2115.7 ± 542.9 ^b^	1881.0 ± 631.7	2132.6 ± 548.9	<0.001
Waist circumference (cm)	81.9 ± 9.2 ^a^	83.0 ± 9.7	84.2 ± 10.1	<0.001	79.6 ± 9.7 ^c^	79.4 ± 9.9	80.5 ± 10.4	<0.001
SBP (mm Hg)	128.4 ± 18.5	129.3 ± 18.2	129.6 ± 20.1	0.123	126.6 ± 20.8 ^c^	126.6 ± 20.6	124.8 ± 21.3	0.001
DBP (mm Hg)	82.0 ± 10.5 ^c^	82.3 ± 10.2	83.3 ± 11.6	<0.001	78.7 ± 10.7 ^a^	79.5 ± 10.6	79.6 ± 11.3	0.008
TG (mmol/L)	1.5 ± 1.7 ^b^	1.9 ± 1.8	1.5 ± 1.7	<0.001	1.4 ± 1.0 ^b^	1.7 ± 1.4	1.4 ± 1.2	<0.001
HDL–C (mmol/L)	1.3 ± 0.4 ^b^	1.3 ± 0.4	1.2 ± 0.3	<0.001	1.2 ± 0.3 ^b^	1.3 ± 0.3	1.3 ± 0.3	<0.001
FPG (mmol/L)	5.0 ± 1.1 ^a^	5.2 ± 1.3	5.1 ± 1.2	<0.001	5.0 ± 1.1 ^a^	5.1 ± 1.1	5.0 ± 1.2	<0.001
Education level				<0.001				<0.001
Primary school or less	889 (44.8) ^a^	713 (35.9)	716 (36.1)		1659 (62.3) ^b^	1369 (51.4)	1452 (54.5)	
Junior high school	810 (40.8)	829 (41.8)	809 (40.8)		794 (29.8)	875 (32.9)	849 (31.9)	
High school and higher	285 (14.4) ^a^	443 (22.3)	460 (23.2)		210 (7.9) ^b^	419 (15.7)	363 (13.6)	
Physical work				<0.001				<0.001
Low physical work	791 (39.9) ^b^	1112 (56.0)	982 (49.5)		1232 (46.3) ^a^	1693 (63.6)	1788 (67.1)	
Middle physical work	198 (10.0) ^a^	236 (11.9)	150 (7.6)		134 (5.0)	178 (6.7) ^c^	85 (3.2)	
High physical work	618 (31.1) ^a^	363 (18.3)	605 (30.5)		985 (37.0) ^a^	483 (18.1)	594 (22.3)	
Other physical work	377 (19.0) ^c^	274 (13.8)	248 (12.5)		312 (11.7) ^c^	309 (11.6)	197 (7.4)	
Region				0.291				<0.001
City	567 (28.6)	612 (30.8)	584 (29.4)		792 (29.7) ^c^	725 (27.2)	966 (36.3)	
Rural	1417 (71.4)	1373 (69.2)	1401 (70.6)		1871 (70.3)	1938 (72.8)	1698 (63.7)	
Geographical region				<0.001				<0.001
Southern Jiangsu	1438 (72.5)	1538 (77.5)	490 (24.7)		1693 (63.6)	1927 (72.4)	664 (24.9)	
Northern Jiangsu	546 (27.5) ^a^	447 (22.5)	1495 (75.3)		970 (36.4) ^a^	736 (27.6)	2000 (75.1)	
Marital status				0.409				<0.001
Unmarried	73 (3.7)	91 (4.6)	100 (5.0)		43 (1.6)	77 (2.9)	88 (3.3)	
Married	1817 (91.6)	1811 (91.2)	1788 (90.1)		2326 (87.3)	2316 (87.0)	2361 (88.6)	
Divorced	13 (0.7)	14 (0.7)	16 (0.8)		17 (0.6)	23 (0.9)	28 (1.1)	
Widowed	81 (4.1)	69 (3.5)	81 (4.1)		277 (10.4) ^a^	247 (9.3)	187 (7.0)	
Smoking behavior				<0.001				<0.001
No	888 (44.8)	926 (46.6)	1019 (51.3)		2590 (97.3)	2612 (98.1)	2619 (98.3)	
Yes	1096 (55.2) ^c^	1059 (53.4)	966 (48.7)		73 (2.7) ^a^	51 (1.9)	45 (1.7)	
Alcohol consumption				0.024				<0.001
Non–drinker	1061 (53.5)	1127 (56.8)	1142 (57.5)		2543 (95.5)	2560 (96.1)	2605 (97.8)	
Current drinker	923 (46.5) ^c^	858 (43.2)	843 (42.5)		120 (4.5) ^c^	103 (3.9)	59 (2.2)	

T: tertile; pattern I: modern dietary pattern; pattern II: vegetable oils/condiments/soy products dietary pattern; pattern III: modern high–wheat pattern dietary pattern; ^a^ vs. other tertiles *p* < 0.05; ^b^ vs. tertile 2 *p* < 0.05; ^c^ vs. tertile 3 *p* < 0.05; BMI, body mass index; SBP, systolic blood pressure; DBP, diastolic blood pressure; TG, triglyceride; TC, total cholesterol; HDL, high–density lipoprotein cholesterol; FPG, fasting plasma glucose.

**Table 4 nutrients-13-04451-t004:** Odds ratios and 95% confidence intervals for metabolic syndrome across tertiles of dietary patterns by genders.

Group	Dietary Pattern	Model 1	Model 2	Model 3
Men	pattern I	OR (95%CI)	OR (95%CI)	OR (95%CI)
	T1	1.000	1.000	1.000
	T2	1.285 (1.105–1.494)	1.161 (0.969–1.389)	1.159 (0.968–1.388)
	T3	1.678 (1.449–1.943)	1.533 (1.273–1.845)	1.530 (1.271–1.842)
	pattern II			
	T1	1.000	1.000	1.000
	T2	0.769 (0.667–0.887)	0.829 (0.700–0.983)	0.829 (0.700–0.983)
	T3	0.692 (0.598–0.800)	0.867 (0.720–1.043)	0.871 (0.723–1.048)
	pattern III			
	T1	1.000	1.000	1.000
	T2	1.462 (1.262–1.695)	1.213 (1.014–1.450)	1.210 (1.012–1.446)
	T3	1.360 (1.172–1.578)	1.162 (0.954–1.416)	1.164 (0.956–1.419)
Women	pattern I			
	T1	1.000	1.000	1.000
	T2	0.922 (0.821–1.035)	1.104 (0.956–1.275)	1.109 (0.960–1.281)
	T3	0.977 (0.871–1.096)	1.280 (1.096–1.493)	1.289 (1.104–1.505)
	pattern II			
	T1	1.000	1.000	1.000
	T2	0.841 (0.750–0.943)	0.872 (0.760–1.001)	0.872 (0.759–1.001)
	T3	0.818 (0.729–0.918)	0.860 (0.744–0.994)	0.863 (0.746–0.997)
	pattern III			
	T1	1.000	1.000	1.000
	T2	1.086 (0.967–1.220)	1.005 (0.873–1.158)	1.006 (0.873–1.158)
	T3	1.121 (0.999–1.259)	1.208 (1.034–1.412)	1.203 (1.030–1.406)

Model 1: unadjusted model; Model 2: adjusted for age, education level, physical work, marital status, region, geographical regions, body mass index, and energy intake; Model 3: additionally adjusted for smoking behavior and alcohol consumption as cluster groups to analyze the association; pattern I: modern dietary pattern; pattern II: vegetable oils/condiments/soy products dietary pattern; pattern III: modern high–wheat pattern dietary pattern.

## Data Availability

Not applicable.

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
