# Peer review of "The Effects of Dietary Pattern on Metabolic Syndrome in Jiangsu Province of China: Based on a Nutrition and Diet Investigation Project in Jiangsu Province"

_nutrients, 2021, doi:10.3390/nu13124451_

Round 1
Reviewer 1 Report
Many thanks for the opportunity to evaluate this manuscript about dietary patterns in a Chinese province of Jiangsu and their associations with metabolic syndrome. The manuscript has large sample size and the results are potentially interesting. However, at its present form the manuscript has a number of issues which should be addressed. Especially introduction and discussion should to be improved and the authors should made it clear how the study is adding to the literature. Please see more detailed comments below.
In places the manuscript would benefit from using more scientific expressions e.g. abstract “figure out”, introduction line 35 “no matter what”, line 47 sentence starting “This is an extremely grim…”.
Abstract
Line 13: „figure out „should be replaced
Abstract should shortly explain what is meant with metabolic syndrome and make clear whether data were used from all the time points mentioned.
Introduction
Line 35: please remove ect. And use more precise language.
Line 56: Reference missing? What is meant with significant effects on health? What effects?
The introduction on the lines 53 to 64 should be expanded and information added about what is meant with evaluation of relationship between diet and disease, why evaluation of dietary patterns is more effective, and what previous studies say (Mediterranean diet is mentioned – but not described). As the authors point out, there is already evidence of potentially healthy dietary patterns – it should be made clear what this study aims to add to the literature.
The definition of metabolic syndrome from line 111 is precise, but should be – shortly – described in the introduction.
Also – did the authors have any hypothesis they tested or was this an explorative study?
Methods
While not clear from the text, it appears that the data was combined for the analysis from three different time points with more than 10 years difference. This should be made clear in the text. It should be also included in the text or the figure how many participants were included from each of the time points. It should also be clarified whether the methodology of data collection was same or similar in all the projects. Furthermore, the authors did not comment whether any sensitivity analyses were done to check that the participants included from the different time points had similar profiles. As authors noted, metabolic syndrome has been increasing – but is there any local evidence whether diet patterns have stayed similar between the time points?
It was also unclear why the patterns were divided in different tertiles.
Statistical methods – please could you clarify whether the factor analysis was done separately for men and women and as the results suggest this might have been the case. How were socio-demographic variables analysed? Please add how the models for the regression were adjusted – this is added below the table but would help to have been included in the methods section.
Results
Line 281: rates – not rats
Table 3 is difficult to follow and while overall significance is given, it is not (clearly) indicated between which tertiles the differences are to be found – and it is still not completely clear why this is important.
Discussion and conclusions
Line 251: Reference, please
Line 258: it should be specified what positive and negative effects in this context mean i.e. negative less risk?
Overall the discussion would benefit from being more concise and better organised, limiting the discussion in the aims of the current study. In the current form the discussion was difficult to follow. The discussion would benefit in concentrating about what new this study to brings to the literature. Considering the observational nature of the study, some of the discussion and conclusion appear very strong.
Further, a number of new references, not included in the introduction, were introduced in the discussion, and made following the discussion difficult. Authors should also avoid including numeric results, whether from this study or other studies. It was also surprising to find the discussion about children, considering that the date regarded adults 18 and over. This should be removed.
Author Response
请参阅附件

Reviewer 2 Report
The authors completed an epidemiological study to determine the association between dietary pattern and metabolic syndrome among adults in Jiangsu province of China. This is a very interesting study, and it would be of interest to the readers of Nutrients. The structure of the paper is good, and most of the necessary information is given. I do have some concerns for the study, which you can find below:
Major comments
The study considers only dietary patterns as factor to be associated with metabolic syndrome. The authors should also consider other factors playing role in the development of metabolic syndrome, such as physical activity. Even though the aim of the study is specific, which is acceptable, the behavioral factor of physical activity should be discussed. Even though the authors collected data regarding the physical work of the participants, this factor was not used in the adjusted models. The authors should at least discuss this in the discussion section.
In the selection of the population to be included in the study, the authors excluded several categories of populations based on specific justifications. Even though this approach sometimes is needed (e.g. presence of disease), some other times is incorporates selection bias (e.g. not complete data). The authors should at least include this in the limitations of their study.
Please specify in the conclusions that these refer to Chinese population.
Minor comments
Please replace throughout the manuscript and abstract the “metabolic syndrome” with “Mets”. This sometimes appears as acronym and sometimes as “metabolic syndrome”.
In sections 3.2 and 3.3 please report the test used for the results including the p values. Even though these appear in table 3, the number of variables is large, and some specification is needed.
Line 281: Please use systematic “review” and meta-analysis, not systematic and meta-analysis
Round 2
Reviewer 1 Report
Thank you for the revision. The authors have attended the points raised and I have no further comments regarding the manuscript.
Reviewer 2 Report
The authors have adequately addressed all my comments